# A Study on the Modifying Effect of Raspberry Seed Oil on Rabbit Meat Quality and Chemical Composition

**DOI:** 10.3390/ani14081150

**Published:** 2024-04-10

**Authors:** Sylwia Ewa Pałka, Zuzanna Siudak, Michał Kmiecik, Łukasz Migdał, Agnieszka Otwinowska-Mindur, Małgorzata Grzesiak

**Affiliations:** 1Department of Genetics, Animal Breeding and Ethology, University of Agriculture in Krakow, Al. Mickiewicza 24/28, 30-059 Kraków, Poland; michal.kmiecik@urk.edu.pl (M.K.); lukasz.migdal@urk.edu.pl (Ł.M.); agnieszka.otwinowska@urk.edu.pl (A.O.-M.); 2Department of Small Livestock Breeding, National Research Institute of Animal Production, Krakowska 1, 32-083 Balice, Poland; zuzanna.siudak@iz.edu.pl; 3Department of Endocrinology, Faculty of Biology, Institute of Zoology and Biomedical Research, Jagiellonian University, Gronostajowa 9, 30-387 Kraków, Poland; m.e.grzesiak@uj.edu.pl

**Keywords:** raspberry seed oil, slaughter and meat quality traits, fatty acid profile, plasma lipid concentration, Termond White rabbits

## Abstract

**Simple Summary:**

We hypothesised a positive effect of adding raspberry seed oil into the feed of purebred Termond White rabbits on slaughter performance traits, plasma lipid concentration and meat quality. In each group, the experimental animals were fed a complete pelleted feed. Rabbits in the first experimental group received a feed ration enriched with a 1% addition of raspberry seed oil, while rabbits in the second experimental group were given a 2% addition of the same oil. The addition of this oil did not significantly affect the slaughter performance traits of the rabbits. It also did not significantly affect the fat content, shear force, hardness, springiness, cohesiveness or chewiness of the meat obtained from the rabbits’ *longissimus lumborum* muscle. However, the addition of raspberry seed oil significantly reduced the levels of tetradecenoic acid and trans-palmitoleic acid. An increase in the linoleic acid content of rabbit meat was also observed. This study showed that feeding pellets containing an oil supplement with a high content of unsaturated fatty acids had a positive effect on rabbits’ plasma lipid levels. The addition of raspberry seed oil might lead to an effective reduction in cholesterol and triglyceride levels in rabbits’ blood.

**Abstract:**

The purpose of the study was to determine the effects of raspberry seed oil on the slaughter performance traits, plasma lipid concentration and meat quality of purebred Termond White rabbits (*n* = 42; 18♂, 24♀). In each group (3 × *n* = 14; 6♂, 8♀), the experimental animals were fed a complete pelleted feed with constant access to drinking water. Rabbits in the first experimental group received a feed ration enriched with a 1% addition of raspberry seed oil, while rabbits in the second experimental group were given a 2% addition of the same oil. These animals were slaughtered on day 84. The addition of raspberry seed oil did not significantly affect the slaughter performance traits of the rabbits (*p* > 0.05). It also did not significantly affect the fat content, shear force, hardness, springiness, cohesiveness or chewiness of the meat obtained from the rabbits’ *longissimus lumborum* muscle. However, the experiment showed that the addition of raspberry seed oil had a significant effect on the fatty acid profile of rabbit meat (*p* ≤ 0.05). Even a small share of this oil in the feed ration significantly increased the level of linoleic acid (*p* ≤ 0.05). This study showed that feeding pellets containing an oil supplement with a high content of unsaturated fatty acids had a positive effect on rabbits’ plasma lipid levels.

## 1. Introduction

Raspberry (*Rubus idaeus* L.) seed oil is characterised by a high content of tocopherol that is found in vitamin E. The vitamin E in this oil is in the form of α-tocopherol and a unique ɣ-tocopherol, which have antioxidant properties, contribute to the delivery of nutrients to cells, and protect erythrocytes from breakdown [1]. Raspberry seed oil also contains vitamin A, carotenoids and flavonoids, which eliminate free radicals. These substances show antiseptic, antibacterial and anti-inflammatory effects [2].

Because raspberry seed oil has a favourable fatty acid profile and a high proportion of acids from the omega-6 group, it can be used as a culinary product. Its addition to a daily diet may satisfy the need for essential fatty acids (EFAs) and vitamin E. In addition, tocopherol in raspberry seed oil may have a positive effect on the immune system, protect the body’s cells from damage caused by free radicals, and influence the health of the nervous system, thus preventing the onset of Alzheimer’s disease and other diseases. Vitamins A and E are responsible for repairing skin cells and soothing inflammation, which suggests that raspberry seed oil can be used in the preparation of skin products with regenerative and anti-ageing effects [2]. The study by Fotschki et al. (2015) [3] conducted on rats revealed that raspberry seed oil had a positive effect on the plasma lipid profile, improved liver function, and alleviated chronic inflammation occurring in rodents.

Based on the above literature review, raspberry seed oil can be used in the cosmetic and medical industries. Its chemical composition, with a high proportion of omega-6 fatty acids and a high natural antioxidant content, suggests that it is worth considering as a nutritional supplement in both human and animal diets [3,4,5,6]. Considering that rabbit meat is a very desirable product to consumers due to its nutritional value, but still lacks consumer acceptance on a large scale, we examined whether the addition of raspberry seed oil in the feed of purebred Termond White rabbits had positive effects on slaughter performance traits, plasma lipid concentration and meat quality.

## 2. Materials and Methods

### 2.1. Animals

The experiment was conducted under standardised conditions at the Experimental Station of the Department of Genetics, Animal Breeding and Ethology of the Agricultural University of Kraków. The experimental animals consisted of 42 rabbits of the purebred Termond White breed (18♂ and 24♀). All of the animals were raised at the Experimental Station of the Department of Genetics, Animal Breeding and Ethology. Until weaning at 35 days of age, young rabbits were kept with their mothers in wooden cages housed in a hall equipped with water (nipple drinkers), lighting (14L:10D) and forced ventilation. From days 35 to 84, the rabbits were kept in metal cages designed for rabbit rearing in the same hall where they stayed with their mothers. The experiment used young rabbits born to 10 does of the basic herd. From each litter, two rabbits were randomly assigned to one of the three study groups (3 × *n* = 14; 6♂, 8♀). In each group, the animals were fed a complete pelleted feed with constant access to drinking water. The rabbits in the first experimental group received feed enriched with a 1% addition of raspberry seed oil (D1), while the rabbits in the second experimental group were given a 2% addition of the same oil (D2). The animals were fed in groups (1 feeder per 2 animals). Cold-pressed raspberry seed oil was bought by the feed manufacturer from an online shop and mixed with all the ingredients that were then pelleted. The component and chemical composition of the feed rations that the rabbits in each group received are presented in Table 1 and Table 2.

### 2.2. Carcass Traits

The animals were slaughtered on day 84 at an average body weight of 2632.75 g ± 51.52 g. The animals were slaughtered after 24 h of starvation, with constant access to drinking water. The animals were stunned, immediately bled, pelted and eviscerated. After slaughter, the carcasses were weighed to determine the warm carcass weight. A further measurement was taken after the carcasses had been chilled for 24 h at 4 °C. The post-slaughter traits included the following: slaughter body weight; hot carcass weight; cold carcass weight; weights of the fore part, middle part and hind part; head weight; and liver weight. All measurements were performed using Łucznik KS-205 electronic scales (Galeria Łucznik Co., Ltd., Wrocław, Poland, e = 0.1). In addition, the hot (DoP1) and cold (DoP2) dressing-out percentages were calculated according to the following formulae: DoP1—carcass weight after slaughter without head and giblets/animal’s body weight before slaughter, expressed as a percentage, and DoP2—cold carcass weight/animal’s body weight before slaughter, expressed as a percentage.

All procedures were performed in accordance with the relevant guidelines and regulations. The authors confirm that the experiment complied with the European Union’s Directive on Animal Experimentation (Directive 2010/63/EU) and ARRIVE guidelines. The experiment was carried out after receiving approval from the 2nd Local Institutional Animal Care and Use Committee (IACUC) in Kraków (agreement no. 267/2018) and the Institutional Animal Care Review Board of the Faculty of Animal Sciences, University of Agriculture in Krakow permissions (approval no. 29/2016, 37/2016, and 3/2018, respectively).

### 2.3. Acidity and Colour Measurement and Fat Content Analysis

At 45 min and 24 h after slaughter, the colour (L*—lightness, a*—redness, and b*—yellowness) and acidity of the rabbit meat were measured. The measurements were performed on the *M. longissimus lumborum* muscle. Meat acidity was measured using a Hanna Instruments HI98163 pH meter (Hanna Instruments Inc., Woonsocket, RI, USA), while meat colour was measured using a Minolta CR-410 colorimeter (Minolta Co., Ltd., Osaka, Japan). The fat content was determined using samples taken from the *M. longissimus lumborum* muscle according to Polish Standards (PN-73/A-82111).

### 2.4. Texture Analysis

Cylindrical meat samples were taken from the right side of the middle part of the carcasses. The samples were vacuum-packed in plastic bags designed for food storage and freezing and then frozen for 72 h at −18 °C. After the stated time, the meat samples were thawed at room temperature and then cooked for 40 min in a water bath set at 80 °C. Shear force and texture profile analysis (TPA) parameters such as hardness, cohesiveness, springiness and chewiness were measured using a TA.XT plus texturometer (Stable Micro Systems Co., Ltd., Godalming, UK). Shear force was measured on the cubic specimens using a Warner–Bratzler blade with a triangular notch. Specimens with a cross-sectional area of 10 × 10 mm were cut perpendicular to the course of the muscle fibres at a blade speed of 2 mm/s. Profile texture analysis was conducted using a cylinder attachment with a 50 mm diameter. A double compression test was performed on the cubic specimens up to 70% of the specimen height. The speed of the roller was 5 mm/s and the interval between pressures was 5 s. All results were counted automatically using Exponent for Windows ver. 5.1.2.0 (Stable Micro Systems Co., Ltd., Godalming, UK), which is compatible with the texturometer software.

### 2.5. Fatty Acids Analysis

The meat samples (*M. longissimus lumborum*) were extracted using a chloroform–methanol solution [7]. Fatty acid methyl esters were prepared according to ISO 12966-2:2011 [8]. The fatty acid profile of the corresponding methyl esters was determined via gas chromatography using a Trace GC Ultra gas chromatograph (Thermo Electron Corp., Waltham, MA, USA) equipped with a Supelcowax 30 m capillary column (Supelco, Bellefonte, PA, USA), with an inner diameter of 0.25 mm and a coating thickness of 0.25 µm. Helium was used as the carrier gas at a flow rate of 1 mL/min. The dispenser and detector temperatures were 220 °C and 250 °C, respectively. The column temperature was initially 160 °C for 3 min and then increased by 3 °C every minute until a temperature of 210 °C was reached, which was maintained for 25 min.

### 2.6. Plasma Cholesterol and Triglycerides Analysis

For further analysis, blood was collected from the animals after slaughter into 1.5 mL heparin-coated tubes. Triglycerides and total cholesterol concentration in the plasma were assessed using commercial colorimetric assays (cat. no T7532 and C7510, respectively; Pointe Scientific, Brussels, Belgium) according to the manufacturer’s protocol. The detection limits were 5 mg/dL for triglycerides and 3 mg/dL for total cholesterol. The intra-assay coefficients of variation for triglycerides and total cholesterol were 1.62% and 1.92%, respectively, and the inter-assay coefficients were 1.32% and 1.93%, respectively. All analyses were performed in duplicate.

### 2.7. Statistical Analysis of the Results

Analysis of variance was performed using the MIXED procedure in SAS [9] to examine post-slaughter traits or meat quality parameters:Y_ijkl_ = µ + Feed_i_ + Sex_j_ + (F × Sex)_ij_ + Sire_k_ + e_ijkl_
where
Y_ijkl_—post-slaughter traits or meat quality parameter,µ—overall mean,Feed_i_—fixed effect of feeding (i = 1, 2, 3),Sex_j_—fixed effect of sex (j = 1, 2),(F × Sex)_ij_—fixed effect of interaction between feeding and sex,Sire_k_—random sire effect (k = 1, 2, 3, 4),e_ijkl_—residual effect.

The significance of differences between means was determined by the Tukey–Kramer test. All *p*-values less than 0.05 were considered as statistically significant.

## 3. Results

### 3.1. Carcass Traits

The addition of raspberry seed oil was not found to significantly affect the slaughter performance traits of the rabbits. However, gender was found to have a significant effect on the rabbits’ slaughter body weight, hot carcass weight, cold carcass weight, weight of the hind part, DoP 1 (hot dressing-out percentage) and DoP 2 (cold dressing-out percentage). The slaughter body weight, hot and cold carcass weights, and weight of the hind part were higher in female Termond White rabbits, but DoP 1 and DoP 2 were higher in male Termond White rabbits (Table 3).

### 3.2. Acidity, Colour Analysis and Fat Content

A significant effect of the addition of raspberry seed oil on the parameter L*_45_ of the *Longissimus lumborum* muscle was observed (Table 4). The highest value was obtained for the meat samples from the animals in the control group, and this value decreased with increasing oil content. Significant differences were noted between the control group and the group with 2% oil (D2). The addition of raspberry seed oil was not found to significantly affect the fat content of the rabbits’ *Longissimus lumborum* muscle (Table 4). In addition, the experiment showed that gender significantly affected the acidity and colour of rabbit meat. The pH measured on the *Longissimus lumborum* muscle after 24 h (Table 4) differed significantly between the groups, with the meat obtained from males having a higher value for this parameter. The results showing the effect of gender on the parameters L*_45_ (lightness), a*_45_ (redness) and b*_45_ (yellowness) of the *longissimus lumborum* muscle are shown in Table 4. In terms of the aforementioned meat colour components, higher values were measured in the meat of female Termond White rabbits. This effect of gender was observed for the a*_24_ and b*_24_ parameters measured after 24 h on this muscle. Gender had no significant effect on the fat content of the *Longissimus lumborum* muscle.

### 3.3. Texture Analysis

The texture analysis showed that the addition of raspberry seed oil, as well as gender, did not significantly affect rabbit meat quality parameters, such as shear force, hardness, springiness, cohesiveness and chewiness (Table 5).

### 3.4. Fatty Acid Profile Analysis

The addition of raspberry seed oil was found to significantly reduce the level of tetradecenoic acid (14:1). Significant differences in the level of this acid were noted between the control group and the two groups that received the addition of raspberry seed oil. The level of trans-palmitoleic acid (16:1 *n*-7) also decreased significantly after the addition of the nutritional supplement. As little as 1% raspberry seed oil supplement (D1) was sufficient for the level of tetradecenoic acid to decrease from 2.30 to 1.38. An increase in the linoleic acid (LA; 18:2 *n*-6) content of rabbit meat was also observed. Significant differences were observed for the content of this acid in the meat from the control animals and those that received a 2% addition of raspberry seed oil (D2) (Table 6). Gender significantly affected the fatty acid levels of rabbit meat. The levels of palmitic acid (16:0), arachidic acid (20:0) and eicosenoic acid (20:1) reached statistically significantly higher values in the meat samples obtained from male rabbits. Anacardic acid (15:0), trans-palmitoleic acid (16:1 *n*-9), margaric acid (17:0), linoleic acid (18:2 *n*-6), gamma-linolenic acid (GLA; 18:3 *n*-6), alpha-linolenic acid (ALA; 18:3 *n*-3) and docosapentaenoic acid (DPA *n*-6; 22:5 *n*-6) were significantly higher in the meat samples from female Termond White rabbits (Table 6).

### 3.5. Plasma Cholesterol and Triglycerides Analysis

Feeding pellets containing an oil supplement with a high content of unsaturated fatty acids had a positive effect on the rabbits’ plasma lipid levels, as shown in Table 7. In addition, significant differences were noted in the plasma triglyceride content between the control group and both experimental groups. Significant differences were also found between the group that received a 1% oil supplement (D1) and the group that received a 2% oil supplement (D2) (Table 7). Moreover, triglyceride levels were significantly influenced by gender, with the levels being significantly lower in the blood of female rabbits (Table 7).

## 4. Discussion

In recent years, we have noticed an increasing interest in adding oils to animal diets. Researchers have attempted to improve the growth performance of rabbits through this strategy [10]; however, the results are ambiguous. The effects of sunflower oil and flaxseed oil enriched with synthetic and natural vitamin E on the slaughter performance traits of rabbits were studied by Eiben et al. (2011) [11]. These researchers found that the groups of animals supplemented with vitamin E oils achieved higher slaughter body weights, and that the carcasses were heavier in animals from the experimental groups compared to the control group. The dressing-out percentage also appeared to be higher in the experimental groups receiving the supplement than in the control group. Matics et al. (2021) [12] determined the effect of silkworm oil on the slaughter performance traits of rabbits. In their study, they replaced sunflower oil used in a commercial complete feed mix with an oil of animal origin. No significant differences were noted in slaughter weight, cold carcass weight, the weight of individual carcass cuts and dressing-out percentage. Our study also showed that the addition of raspberry seed oil did not have a significant effect on the slaughter performance traits of rabbits. El-Hack et al. (2018) [13] studied the effects of black and red pepper oil on the slaughter performance traits of New Zealand White rabbits and found that the oil supplement did not affect parameters such as dressing-out percentage, carcass weight, liver weight, spleen weight, heart weight and lung weight. However, it significantly decreased kidney weight in the studied animals. Additionally, their study found an increase in dressing-out percentage with an increase in the proportion of pepper oil in the feed ration fed to rabbits. In our study, the values of this parameter were not only significantly higher in the experimental groups, but also the relationship between oil content in the ration and dressing-out percentage was inverse.

The results of our study indicate that gender significantly influences post-slaughter traits in rabbits. Pałka et al. (2016) [14] also showed that all of the abovementioned traits were higher in females, but the differences were not significant. In contrast, the higher values of hot and cold dressing-out percentages were characteristic of carcasses obtained from males. Ortiz Hernandez and Rubio Lozano (2001) [15] studied the effect of the sex of rabbits of meat breeds on dressing-out percentages and, although they did not observe significant differences, carcasses from males were characterised by a higher value of these parameters, as in our experiment.

Cullere et al. (2018) [16] examined the effect of flaxseed additive—an oilseed crop—on the quality of rabbit meat. At 24 h after slaughter, the first analysis of the acidity and colour of the meat samples was carried out. Their experiment showed that the addition of oilseeds did not significantly affect the acidity and colour of rabbit meat 24 h after slaughter. The average values obtained in their experiment differed significantly from the results obtained in our in-house experiment. The differences might be due to the use of Pannon White rabbits in the experiment conducted by Cullere et al. (2018) [16], and the fact that acidity and colour were measured on loin samples isolated during dissection, whereas the analysis in our experiment was carried out on the whole carcasses. As mentioned in the Section 1, raspberry seed oil is rich in vitamin E (tocopherol) and other antioxidants, which have a positive effect on meat stability and subsequent shelf life. Virag et al. (2008) [17] investigated how vitamin E supplementation affected meat quality traits such as colour and acidity measured 24 h after slaughter. In that study, the values of all meat quality parameters examined were found to be higher in the meat samples from animals supplemented with the natural form of vitamin E, with significant differences found between the means of the parameters L* and a* measured on the *Longissimus lumborum* muscle. The meat of rabbits given a dietary ration with a higher vitamin E content had a higher pH value compared to rabbits given a feed ration that had a lower vitamin E content. Eiben et al. (2010) [18] reported that the use of sunflower and linseed oil blends in rabbit diets had a lowering effect on meat colour parameters. However, there was no effect of the additives on the acidity of rabbit meat. In our experiment, no significant effects of the oil additive on meat colour and pH were found. However, in the study by Eiben et al. (2010) [18], a higher proportion of the tested oil (4%) was used.

In our study, we did not find an effect of nutritional supplementation and gender on the texture parameters of rabbit meat. It is possible that an experiment in which a higher proportion of raspberry seed oil is tested would have produced more satisfactory results, showing the effect of the oil on meat texture characteristics. The results reported by Pałka et al. (2021) [19] indicated that a nutritional supplement with herbs, such as nettle and fenugreek, had no significant effect on meat texture characteristics.

Despite the lack of significant differences, it could be observed that the meat obtained from female rabbits was characterised by higher values for all the tested meat texture parameters. Female rabbits of meat breeds mature faster than males, which is associated with the deposition of intramuscular fat; this affects meat quality by increasing its tenderness and tastiness, but it may also affect meat texture parameters [20].

Rabbit meat is a product with many health-promoting properties. This is due to its very favourable fatty acid profile, among other benefits. Papadomichelakis et al. (2010) [21] tested the effect of 20% soybean oil on the fatty acid profile of rabbit meat and found that the addition of this oil had a significant effect mainly on unsaturated fatty acids. Soybean oil had no effect on long-chain polyunsaturated fatty acids, but it had a positive effect on the ratio of saturated to unsaturated acids. As in our experiment, the addition of soybean oil increased the content of linoleic acid (LA; cis-18:2 *n*-6). The addition of soybean oil also significantly decreased the content of trans-palmitoleic acid (trans-7-C16:1). Tres et al. (2009) [22] compared the effect of an oil rich in *n*-3 acids (linseed oil) with an oil rich in *n*-6 acids (sunflower oil). Their experiment showed that sunflower oil increased the content of *n*-6-bonded acids in rabbit meat. Among other things, significant differences were observed in the level of linoleic acid. Likewise, raspberry seed oil predominantly contains *n*-6 forms of acids and, therefore, does not affect acids such as ALA, EPA or DHA.

The effect of gender on the fatty acid profile was investigated by North et al. (2019) [23]. Their study showed that gender significantly affected the level of stearic acid in rabbit meat, with the meat obtained from males having a higher content of this saturated acid. Gender, as in our experiment, also significantly affected the level of linoleic acid (LA; cis-18:2n-6). However, in our study, the meat from female rabbits was characterised by a higher concentration of linoleic acid, while in the experiment by North et al. (2019) [23], a higher level of linoleic acid was found in the meat from male rabbits, with a concentration of only 13.6. New Zealand White rabbits were used in their study, and breed may also be a factor influencing the fatty acid profile of rabbit meat, leading to the differences observed.

Our experiment demonstrated the beneficial effects of raspberry seed oil, a small amount of which was able to reduce the plasma triglyceride levels by up to half and reduce total cholesterol levels. Although this is the first research study on the effects of raspberry seed oil conducted in rabbits, similar results were obtained in studies utilising other rodent species. In rats and Syrian hamsters, raspberry seed oil markedly decreased triglyceride plasma concentration [3,24]. Raspberry seed oil is rich in polyunsaturated fatty acids (PUFAs), which are known to inhibit hepatic lipase activity and very-low-density lipoprotein synthesis [4]; this might explain the results obtained in this study. Furthermore, the study by Lee et al. (2018) [5] revealed a positive effect of black raspberry seed oil on lipid metabolism in obese mice, suggesting that this oil might inhibit lipogenesis and promote fatty acid oxidation. On the other hand, Pieszka et al. (2013) [6] did not observe any significant effect of raspberry seed oil on the blood lipid profile of rats, which might be due to different doses of supplementation or species differences.

## 5. Conclusions

Based on the analyses described above, it can be concluded that the addition of raspberry seed oil does not have a negative effect on the slaughter performance traits and meat quality of purebred Termond White rabbits. However, a small share of this oil in the feed ration can significantly increase the level of linoleic acid, which builds cells and serves as an important substrate in the biosynthesis of compounds affecting the functioning of organisms. Furthermore, the addition of this oil might lead to an effective reduction in cholesterol and triglyceride levels in the blood of rabbits.

## Figures and Tables

**Table 1 animals-14-01150-t001:** Ingredients and chemical composition of the control and experimental feed (according to feed manufacturer FHP Barbara Ltd., Turza, Poland).

Component	Feeding ^†^		
	C	D1	D2
Ingredients (%)			
Wheat	29.58	28.58	27.58
Maize	24.50	24.50	24.50
Bran	15.00	15.00	15.00
Sunflower meal	11.00	11.00	11.00
Lucerne meal	10.00	10.00	10.00
Soya meal	7.00	7.00	7.00
Mineral and vitamin premix	1.50	1.50	1.50
Calcium carbonate	0.80	0.80	0.80
Dicalcium phosphate	0.62	0.62	0.62
Raspberry seed oil	0.00	1.00	2.00
Chemical composition (%)			
Dry matter	89.07	89.12	89.03
Crude ash	7.40	7.27	7.39
Total nitrogen	2.57	2.54	2.50
Total protein	16.42	15.96	16.00
Crude fat	3.90	4.02	4.11
Crude fibre	14.62	14.39	14.96
Metabolic energy [MJ/kg]	20.20	19.60	19.40
Dry matter	89.07	89.12	89.03

^†^ C—control group, D1—feed with 1% raspberry seed oil added, D2—feed with 2% raspberry seed oil added.

**Table 2 animals-14-01150-t002:** Fatty acid profile of feed mixtures fed to rabbits in the control and experimental groups.

Fatty Acid	Feeding ^†^			
	C	D1	D2	RSO
10:0 (decanoic acid)	0.72	0.34	0.39	0.00
12:0 (lauric acid)	10.56	4.92	5.44	0.00
14:0 (myristic acid)	3.90	1.75	1.99	0.09
14:1 (myristoleic acid)	0.03	0.04	0.04	0.00
15:0 (pentadecanoic acid)	0.12	0.08	0.09	0.02
16:0 (palmitic acid)	12.87	11.12	12.55	6.75
16:1 *n*-9 (palmitoleic acid)	0.07	0.13	0.11	0.01
16:1 *n*-7 (palmitoleic acid)	0.26	0.20	0.19	0.13
17:0 (heptadecanoic acid)	0.09	0.08	0.09	0.04
17:1 (heptadecenoic acid)	0.06	0.05	0.04	0.03
18:0 (stearic acid)	3.42	2.87	3.55	3.71
18:1 *n*-9 (oleic acid)	29.76	30.26	27.98	24.88
18:1 *n*-7 (cis-Vaccenic acid)	1.10	1.13	0.97	0.53
18:2 *n*-6 (linoleic acid)	32.90	43.42	43.20	60.15
18:3 *n*-6 (gamma-linolenic acid)	0.01	0.01	0.02	0.02
18:3 *n*-3 (a-linolenic acid)	3.41	2.93	2.65	3.23
20:0 (arachidic acid)	0.30	0.33	0.42	0.29
20:1 (eicosenoic acid)	0.44	0.36	0.31	0.12
Total	100.02	100.02	100.03	100.00

^†^ C—control group, D1—feed with 1% raspberry seed oil added, D2—feed with 2% raspberry seed oil added, RSO—raspberry seed oil.

**Table 3 animals-14-01150-t003:** Effect of feeding and gender on slaughter performance traits.

Parameter	Feeding ^†^							Gender				
	C *n* = 14		D1 *n* = 14		D2 *n* = 14		*p*-Value	♂ *n* = 18		♀ *n* = 24		*p*-Value
	LSM ^‡^	SE ^§^	LSM ^‡^	SE ^§^	LSM ^‡^	SE ^§^		LSM ^‡^	SE ^§^	LSM ^‡^	SE ^§^	
Slaughter weight [g]	2656.32	52.62	2660.29	51.47	2581.63	50.58	0.4095	2504.28 ^a^	47.34	2761.22 ^b^	40.65	<0.0001
Hot carcass weight [g]	1425.57	31.84	1429.82	31.26	1353.50	30.81	0.1219	1353.72 ^a^	28.51	1452.20 ^b^	24.51	0.0053
Cold carcass weight [g]	1359.30	30.34	1361.76	29.84	1291.88	29.46	0.1426	1292.83 ^a^	27.10	1382.47 ^b^	23.32	0.0081
Fore part [g]	560.79	16.47	547.58	16.47	534.00	16.47	0.5224	534.33	14.38	560.58	12.45	0.1763
Middle part [g]	289.17	16.97	304.27	16.91	277.34	16.86	0.4940	275.34	14.89	311.19	12.88	0.0706
Hind part [g]	488.00	9.39	498.79	9.39	470.46	9.39	0.1134	471.00 ^a^	8.20	500.50 ^b^	7.10	0.0101
Head [g]	128.18	2.39	132.23	2.39	130.98	2.39	0.4767	130.17	2.09	130.75	1.80	0.8339
Liver [g]	69.43	2.68	65.35	2.61	68.09	2.56	0.4514	67.93	2.42	67.31	2.08	0.8182
DoP1 ^¶^ [%]	54.00	0.45	53.97	0.45	52.64	0.45	0.0613	54.32 ^a^	0.39	52.76 ^b^	0.34	0.0047
DoP2 ^¶^ [%]	51.60	0.44	51.50	0.44	50.34	0.44	0.0924	51.99 ^a^	0.38	50.31 ^b^	0.33	0.0022

^†^ C—control group, D1—feed with 1% raspberry seed oil added, D2—feed with 2% raspberry seed oil added; ^‡^ LSM—least square means; ^§^ SE—standard error; ^¶^ DoP1—hot dressing out percentage; DoP2—cold dressing out percentage. ^a, b^—averages marked with different letters are significantly different (*p* ≤ 0.05).

**Table 4 animals-14-01150-t004:** Effect of feeding and gender on meat acidity, color and total fat—*longissimus lumborum* muscle.

Parameter	Feeding ^†^							Gender				
	C *n* = 14		D1 *n* = 14		D2 *n* = 14		*p*-Value	♂ *n* = 18		♀ *n* = 24		*p*-Value
	LSM ^‡^	SE ^§^	LSM ^‡^	SE ^§^	LSM ^‡^	SE ^§^		LSM ^‡^	SE ^§^	LSM ^‡^	SE ^§^	
pH_45_	6.27	0.07	6.48	0.07	6.32	0.07	0.0598	6.35	0.06	6.34	0.05	0.9743
pH_24_	5.07	0.03	5.08	0.03	5.09	0.03	0.8967	5.15 ^a^	0.03	5.01 ^b^	0.02	<0.0001
L*_45_	55.77 ^b^	0.94	54.95 ^ab^	0.94	52.12 ^a^	0.94	0.0243	52.19 ^a^	0.82	56.38 ^b^	0.71	0.0005
a*_45_	4.47	0.54	4.93	0.54	3.79	0.53	0.2781	3.15 ^a^	0.49	5.65 ^b^	0.42	0.0001
b*_45_	0.93	0.40	1.16	0.40	0.31	0.40	0.3129	−0.26 ^a^	0.35	1.86 ^b^	0.31	<0.0001
L*_24_	57.06	0.61	57.27	0.59	57.33	0.58	0.9335	57.10	0.54	57.33	0.47	0.7154
a*_24_	9.57	0.60	8.56	0.60	8.81	0.60	0.4645	7.97 ^a^	0.52	9.98 ^b^	0.45	0.0060
b*_24_	5.56	0.39	5.60	0.39	6.06	0.39	0.6191	4.60 ^a^	0.34	6.88 ^b^	0.30	<0.0001
Fat content [%]	2.37	0.60	2.46	0.59	2.47	0.60	0.4520	2.40	0.49	2.46	0.49	0.3598

^†^ C—control group, D1—feed with 1% raspberry seed oil added, D2—feed with 2% raspberry seed oil added; ^‡^ LSM—least square means; ^§^ SE—standard error. ^a, b^—averages marked with different letters are significantly different (*p* ≤ 0.05).

**Table 5 animals-14-01150-t005:** Effect of feeding and gender on shear force and profile texture analysis of rabbit meat.

Parameter	Feeding ^†^							Gender				
	C *n* = 14		D1 *n* = 14		D2 *n* = 14		*p*-Value	♂ *n* = 18		♀ *n* = 24		*p*-Value
	LSM ^‡^	SE ^§^	LSM ^‡^	SE ^§^	LSM ^‡^	SE ^§^		LSM ^‡^	SE ^§^	LSM ^‡^	SE ^§^	
Shear force [kg]	1.47	0.16	1.43	0.15	1.70	0.15	0.3399	1.42	0.13	1.65	0.13	0.1520
Hardness [kg]	12.65	0.94	11.25	0.94	11.93	0.94	0.5820	11.43	0.76	12.46	0.76	0.3517
Springiness	0.57	0.03	0.53	0.03	0.55	0.03	0.5613	0.52	0.02	0.58	0.02	0.0614
Cohesiveness	0.48	0.02	0.45	0.02	0.46	0.02	0.4795	0.45	0.02	0.47	0.02	0.2193
Chewiness [kg]	3.66	0.45	2.86	0.45	3.22	0.45	0.4630	2.86	0.36	3.65	0.36	0.1549

^†^ C—control group, D1—feed with 1% raspberry seed oil added, D2—feed with 2% raspberry seed oil added; ^‡^ LSM—least square means; ^§^ SE—standard error.

**Table 6 animals-14-01150-t006:** Effect of feeding and gender on the fatty acid profile of rabbit meat.

Fatty Acid	Feeding ^†^							Gender				
	C *n* = 14		D1 *n* = 14		D2 *n* = 14		*p*-Value	♂ *n* = 18		♀ *n* = 24		*p*-Value
	LSM ^‡^	SE ^§^	LSM ^‡^	SE ^§^	LSM ^‡^	SE ^§^		LSM ^‡^	SE ^§^	LSM ^‡^	SE ^§^	
10:0	0.06	0.01	0.06	0.01	0.05	0.01	0.8805	0.05	0.01	0.06	0.01	0.2406
12:0	0.65	0.05	0.64	0.05	0.70	0.05	0.6056	0.67	0.04	0.65	0.04	0.6196
14:0	2.62	0.11	2.40	0.11	2.64	0.10	0.1691	2.57	0.09	2.53	0.09	0.7184
14:1	0.15 ^b^	0.02	0.08 ^a^	0.02	0.07 ^a^	0.02	0.0050	0.09	0.02	0.11	0.02	0.3520
15:0	0.48	0.02	0.51	0.02	0.50	0.02	0.5631	0.46 ^a^	0.01	0.53 ^b^	0.01	0.0016
16:0	25.99	0.60	24.79	0.60	24.58	0.60	0.2354	26.94 ^a^	0.49	23.30 ^b^	0.49	<0.0001
16:1 *n*-9	0.47	0.02	0.51	0.02	0.48	0.02	0.2604	0.46 ^a^	0.02	0.52 ^b^	0.02	0.0196
16:1 *n*-7	2.30 ^b^	0.21	1.38 ^a^	0.21	1.39 ^a^	0.20	0.0041	1.75	0.18	1.63	0.18	0.6080
17:0	0.54	0.02	0.58	0.02	0.56	0.02	0.2244	0.53 ^a^	0.01	0.59 ^b^	0.01	0.0068
17:1	0.26	0.02	0.22	0.02	0.21	0.02	0.0658	0.22	0.01	0.24	0.01	0.1737
18:0	6.20	0.29	6.58	0.29	6.54	0.28	0.7149	6.49	0.24	6.40	0.24	0.7571
18:1 *n*-9	28.95	0.48	28.27	0.47	29.33	0.46	0.2535	28.87	0.40	28.82	0.40	0.9230
18:1 *n*-7	1.28	0.06	1.30	0.06	1.20	0.06	0.4171	1.22	0.05	1.29	0.05	0.2909
18:2 *n*-6	23.12 ^b^	0.53	24.80 ^a b^	0.53	25.41 ^a^	0.53	0.0188	23.48 ^a^	0.43	25.41 ^b^	0.43	0.0056
18:3 *n*-6	0.06	0.01	0.07	0.01	0.06	0.01	0.4607	0.05 ^a^	0.01	0.07 ^b^	0.01	0.0175
18:3 *n*-3	1.52	0.08	1.52	0.08	1.66	0.08	0.3271	1.45 ^a^	0.06	1.68 ^b^	0.06	0.0129
CLA	0.03	0.01	0.03	0.01	0.02	0.01	0.2658	0.02	0.01	0.03	0.01	0.0936
20:0	0.11	0.01	0.12	0.01	0.12	0.01	0.6646	0.13 ^a^	0.01	0.10 ^b^	0.01	0.0220
20:1	0.36	0.02	0.35	0.02	0.35	0.02	0.7874	0.39 ^a^	0.01	0.32 ^b^	0.01	0.0012
20:2	0.28	0.02	0.28	0.02	0.25	0.02	0.4599	0.25	0.01	0.29	0.01	0.0650
20:3 *n*-6	0.27	0.03	0.28	0.03	0.21	0.03	0.2490	0.22	0.03	0.29	0.03	0.0637
20:4 *n*-6	2.63	0.50	3.25	0.49	2.22	0.47	0.2877	2.26	0.41	3.14	0.41	0.1063
20:4 *n*-3	0.04	0.01	0.04	0.01	0.03	0.01	0.5254	0.03	0.01	0.04	0.01	0.0712
20:5 *n*-3	0.06	0.01	0.06	0.01	0.04	0.01	0.3134	0.04	0.01	0.06	0.01	0.0808
22:4 *n*-6	0.84	0.13	1.03	0.13	0.73	0.13	0.2543	0.73	0.11	1.01	0.11	0.0714
22:5 *n*-6	0.34	0.05	0.40	0.05	0.26	0.05	0.3009	0.24 ^a^	0.04	0.39 ^b^	0.04	0.0158
22:5 *n*-3	0.33	0.05	0.39	0.05	0.28	0.05	0.3317	0.29	0.05	0.38	0.05	0.1219
22:6 *n*-3	0.09	0.01	0.11	0.01	0.08	0.01	0.3032	0.08	0.01	0.11	0.01	0.0668
Total	99.97	0.01	99.97	0.01	99.98	0.01	0.3810	99.98	0.01	99.97	0.01	0.0891

^†^ C—control group, D1—feed with 1% raspberry seed oil added, D2—feed with 2% raspberry seed oil added; ^‡^ LSM—least square means; § SE—standard error. ^a, b^—averages marked with different letters are significantly different (*p* ≤ 0.05).

**Table 7 animals-14-01150-t007:** Effect of type of feeding on blood lipidogram of rabbits.

Parameter	Feeding ^†^							Gender				
	C *n* = 14		D1 *n* = 14		D2 *n* = 14		*p*-Value	♂ *n* = 18		♀ *n* = 24		*p*-Value
	LSM ^‡^	SE ^§^	LSM ^‡^	SE ^§^	LSM ^‡^	SE ^§^		LSM ^‡^	SE ^§^	LSM ^‡^	SE ^§^	
Cholesterol [mg/dL]	2220.75 ^b^	54.61	1197.75 ^a^	54.61	1013.42 ^a^	54.61	<0.0001	1494.11	44.59	1460.50	44.59	0.0011
Triglycerides [mg/dL]	663.00 ^c^	12.59	501.83 ^a^	12.59	377.33 ^b^	12.59	<0.0001	541.22 ^a^	10.28	486.89 ^b^	10.28	<0.0001

^†^ C—control group, D1—feed with 1% raspberry seed oil added, D2—feed with 2% raspberry seed oil added; ^‡^ LSM—least square means; ^§^ SE—standard error. ^a, b, c^—averages marked with different letters are significantly different (*p* ≤ 0.05).

## Data Availability

Data are contained within the article.

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
