# Peer review of "A Study on the Modifying Effect of Raspberry Seed Oil on Rabbit Meat Quality and Chemical Composition"

_animals, 2024, doi:10.3390/ani14081150_

Round 1
Reviewer 1 Report
Comments and Suggestions for Authors
The aim of the study was to determine the effect of raspberry seed oil (Rubus idaeus L.) on the carcass characteristics of Termond White rabbits and on blood lipid concentration. Forty-two experimental animals were divided into three equal groups (n = 14; 6 males, 8 females), with groups D1 and D2 receiving 1% and 2% raspberry seed oil supplements, respectively, added to the feed provided.
The work represents a typical nutritional experiment, with the innovation being the use of raspberry seed oil as a supplement to the basic feed ration. The experiment showed a beneficial effect of raspberry seed oil on the levels of triglycerides and total cholesterol in the blood, resulting in their reduction. This could serve as a starting point for further research of this kind, especially considering the promising preliminary results. I recommend publishing the work after addressing the following comments.
Comments:
• When indicating the number of experimental groups, please maintain a consistent order regarding gender designation throughout the text, e.g., total number, males, females. In Verse 3, it is written as (n=42; 18, 24), further (n=14; 8, 6). Change it to (n=42; 18, 24), further (n=14; 6, 8).
• Were the animals transferred to new cages and grouped into breeding sets before the experiment began? If so, was an acclimatization period to the new conditions considered? It would be worthwhile to mention this in the methodology.
• How was the oil used in the experiment obtained? Information on this would be important, as the method of oil extraction (cold-pressed, hot-pressed) affects the nutritional value of the product and the content of certain vitamins, among other factors.
• During what period was the blood collected, and was it collected by an authorized person?
• The incorrectly entered number of females in the tables is: 124, it should be 24.
Author Response
Dear Reviewer,
thank you for all your comments. We have made corrections to the text of the manuscript. At the same time, we would like to inform you that the entire text has been checked and corrected by the MDPI translators.
- When indicating the number of experimental groups, please maintain a consistent order regarding gender designation throughout the text, e.g., total number, males, females. In Verse 3, it is written as (n=42; 18, 24), further (n=14; 8, 6). Change it to (n=42; 18, 24), further (n=14; 6, 8).
Answer: In the abstract and in the chapter entitled Material and methods, the record concerning the number of female and male rabbits has been corrected.
- Were the animals transferred to new cages and grouped into breeding sets before the experiment began? If so, was an acclimatization period to the new conditions considered? It would be worthwhile to mention this in the methodology.
Answer: The experimental animals were transferred to metal cages at day 35. The cages are located exactly in the same hall in which the animals were housed with their mothers. For this reason, we did not carry out acclimatisation. Instead, in order to protect the animals from pathogens, the cages and their equipment were thoroughly disinfected beforehand.
- How was the oil used in the experiment obtained? Information on this would be important, as the method of oil extraction (cold-pressed, hot-pressed) affects the nutritional value of the product and the content of certain vitamins, among other factors.
Answer: The oil used for the experiment was cold-pressed. Missing information in the text of the manuscript has been completed.
- During what period was the blood collected, and was it collected by an authorized person?
Answer: Blood was collected from animals post-slaughter. Missing information is included in the text of the manuscript.
The incorrectly entered number of females in the tables is: 124, it should be 24.
Answer: The number of females in the tables has been corrected.
Reviewer 2 Report
Comments and Suggestions for Authors
This paper, titled The study on the modifying effect of Raspberry seed oil upon the rabbits' slaughter meat productivity, deals with a very interesting topic, namely the use of an alternative feed supplement on the carcass characteristics of rabbit meat. Nowadays, when the quality and nutritional value of the meat consumed by the consumer is very important, finding ways to improve the quality of the meat is important.
This manuscript evaluates an interesting feed supplement that is relatively new which is a great added value. However, I have some major comments on this manuscript that should be clarified.
1. The animal being studied is a Termond White rabbit - is it a hybrid or purebred? In the Introduction it would be useful to mention if there are any differences between breeds or hybrids. Alternatively, characterise the breed in the methodology.
2. In the methodology you list a total of 42 rabbits - but no gender is mentioned.
3. The rabbits were divided into 3 groups - In each group (n = 14; 8, 6) - why are the numbers given here different than in the tables (K 14, D1 14, D2 14)?
4. In Tables 3 - 6 the control group is labelled K but in the footnotes it is listed as C.
5. The study also evaluates the effect of gender on each parameter. I assume that the animals were 18 and 24 is stated in the methodology. But in the tables the number of females is given as 124.
6. The number of males and females is given only in the tables but not in the methodology. In addition, the gender distribution in each dietary group is not mentioned.
7. Was the effect of gender taken into account when evaluating the effect of the addition of raspberry seed oil? This would be useful for the validity of the results.
Before publishing the manuscript, these ambiguities should be clarified and completed.
Author Response
Dear Reviewer,
thank you for all your comments. We have made corrections to the text of the manuscript. At the same time, we would like to inform you that the entire text has been checked and corrected by the MDPI translators.
- The animal being studied is a Termond White rabbit - is it a hybrid or purebred? In the Introduction it would be useful to mention if there are any differences between breeds or hybrids. Alternatively, characterise the breed in the methodology.
Answer: The rabbits used in the experiment were purebred animals. The Termond White breed is a very popular medium breed of rabbit. It is characterized by a fast growth rate and high meat quality. In addition, females are characterized by high fertility and good maternal qualities.
Many scientists have confirmed the advantages of this breed, so we believe that in a paper on nutrition rather than a comparison of several breeds of rabbits, it is not necessary to characterize this breed in the Introduction or Material and Methods chapters.
- In the methodology you list a total of 42 rabbits - but no gender is mentioned.
Answer: The number of males and females used in the experiment is included in the Abstract, in the section titled Material and methods, and in the tables.
- The rabbits were divided into 3 groups - In each group (n = 14; 8, 6) - why are the numbers given here different than in the tables (K 14, D1 14, D2 14)?
Answer: 42 animals were used for the experiment (line 76). The animals were divided into three groups: control group (C; n=14), first experimental group (D1; n=14) and second experimental group (D2; n=14)
- In Tables 3 - 6 the control group is labelled K but in the footnotes it is listed as C.
Answer: We have corrected the designation of the control group in the tables. In each table, the control group is denoted by the letter C.
- The study also evaluates the effect of gender on each parameter. I assume that the animals were 18 and 24 is stated in the methodology. But in the tables the number of females is given as 124.
Answer: The number of females in the tables has been corrected.
- The number of males and females is given only in the tables but not in the methodology. In addition, the gender distribution in each dietary group is not mentioned.
Answer: A description of the number of females and males from each group can be found in the line 84.
- Was the effect of gender taken into account when evaluating the effect of the addition of raspberry seed oil? This would be useful for the validity of the results.
Answer: A detailed description of the effect of gender on the examined parameters can be found in the chapter entitled Results in lines 190-195, 209-216, 219-221, 232-238, 247-249.
Reviewer 3 Report
Comments and Suggestions for Authors
The present study by Palka et al evaluated the effect of the raspberry seed dietary supplementation in the rabbit diet. The study has some interesting findings and may be useful for cuniculture. However, I have the following observations-
As rabbit meat is already rich in PUFA and has a desirable lipid profile, so authors need to add some point to strengthen the hypothesis.
The abstract does not reflect the quantity and quality of the work. I believe the authors have conducted an exhaustive study with a good sample size to conclude the findings, but not clearly reflected it in abstract.
i. L3: Please check the experiment's n value; it is not clear, and it seems the author used a different sample size for different feed trials. Is it refer male and female?
ii. L7: level of significance
iii. In abstract, please use also summarize the findings of other traits (if non-significant, then also please mention) meat quality
iv. Keywords: Have scope for improvement
v. L 14: please add the missing form of tocopherol
vi. L32: desirable due to nutritive value, but still lacks consumer acceptance on large scale
vii. May also add previous work on feeding of raspberry seed on livestock growth and meat quality to strengthen your hypothesis; as it was mentioned in discussion section.
viii. L57: is the feed withdrawal period is as per regulations as 24 h seems too much time. May be due to cecotrophy?
ix. Also mention the slaughtering process and mention the ethical approval with approved date. (Although mention at the end of the manuscript)
x. L108: time of blood collection, during slaughter or during feeding, please make it clear
xi. Please italic the muscle name throughout the manuscript
xii. In Tables: please explain the n=124 for females, when total rabbits are 42 only,
Author Response
Dear Reviewer,
thank you for all your comments. We have made corrections to the text of the manuscript. At the same time, we would like to inform you that the entire text has been checked and corrected by the MDPI translators.
The abstract does not reflect the quantity and quality of the work. I believe the authors have conducted an exhaustive study with a good sample size to conclude the findings, but not clearly reflected it in abstract.
Answer: We have improved the abstract (line 32-40).
- L3: Please check the experiment's n value; it is not clear, and it seems the author used a different sample size for different feed trials. Is it refer male and female?
Answer: During the preparation of our manuscript for the requirements of the journal, we made the mistake of entering the number of females incorrectly. We have corrected our error in the abstract and tables.
- L7: level of significance
Answer: A level of significance was added (line 34, 37, 38).
iii. In abstract, please use also summarize the findings of other traits (if non-significant, then also please mention) meat quality
Answer: The abstract has been corrected. We have included information on the effect of raspberry seed oil on slaughter characteristics and meat quality (line 33-40).
- Keywords: Have scope for improvement
Answer: The keywords have been changed (line 41-42).
- L 14: please add the missing form of tocopherol
Answer: missing forms of tocopherol added (line 46 and 47).
- L32: desirable due to nutritive value, but still lacks consumer acceptance on large scale
Answer: we added the text (line 66).
vii. May also add previous work on feeding of raspberry seed on livestock growth and meat quality to strengthen your hypothesis; as it was mentioned in discussion section.
Answer: the publications on which the hypothesis was based are cited (line 65).
viii. L57: is the feed withdrawal period is as per regulations as 24 h seems too much time. May be due to cecotrophy?
Answer: A 24-hour starvation period for rabbits is the standard procedure used in Poland. This time makes the slaughter performance results most accurate. In addition, thanks to such starvation, less slaughter waste is left for utilization after slaughter.
Below are some publications that provide confirmation of the methodology we used in our experiment.
View of THE EFFECTS OF PRESLAUGHTER WITHHOLDING OF FEED AND WATER FROM RABBITS ON THEIR CARCASS YIELD AND MEAT QUALITY, (njap.org.ng)
View of The effect of intensive and extensive production systems on carcass quality in New Zealand White rabbits (upv.es)
Evaluation-of-Slaughter-Parameters-and-Meat-Quality-of-Rabbits-Fed-Diets-with-Silkworm-Pupae-and-Mealworm-Larvae-Meals.pdf
The effect of transport on the quality of rabbit meat (wiley.com)
View of Growth performance and meat composition of rabbits fed diets supplemented with silkworm pupae meal (csic.es)
- Also mention the slaughtering process and mention the ethical approval with approved date. (Although mention at the end of the manuscript)
Answer: the missing information has been added (line 116-123).
L108: time of blood collection, during slaughter or during feeding, please make it clear
Answer: Blood was collected from animals post-slaughter. Missing information is included in the text of the manuscript (line 164).
Please italic the muscle name throughout the manuscript
Answer: The notation of muscle names has been corrected. They are now written in italics.
xii. In Tables: please explain the n=124 for females, when total rabbits are 42 only,
Answer: The number of females in the tables has been corrected.
Round 2
Reviewer 2 Report
Comments and Suggestions for Authors
In my opinion, all the questions have been answered and the necessary information has been added. The authors have done a good job of incorporating the comments. The manuscript can be published.